# Biodegradation of Selected Hydrocarbons by *Fusarium* Species Isolated from Contaminated Soil Samples in Riyadh, Saudi Arabia

**DOI:** 10.3390/jof9020216

**Published:** 2023-02-06

**Authors:** Fatimah Al-Otibi, Rasha M. Al-Zahrani, Najat Marraiki

**Affiliations:** Department of Botany and Microbiology, College of Science, King Saud University, P.O. Box 22452, Riyadh 11495, Saudi Arabia

**Keywords:** hydrocarbons biodegradation, *Fusarium* spp., biosurfactants, DCPIP, oil-emulsification, drop-collapse, germination-assay

## Abstract

Background: Microbial biodegradation of oil-hydrocarbons is one of the sustainable and cost-effective methods to remove petroleum spills from contaminated environments. The current study aimed to investigate the biodegradation abilities of three *Fusarium* isolates from oil reservoirs in Saudi Arabia. The novelty of the current work is that the biodegradation ability of these isolates was never tested against some natural hydrocarbons of variable compositions, such as Crude oil, and those of known components such as kerosene and diesel oils. Methods: The isolates were treated with five selected hydrocarbons. The hydrocarbon tolerance test in solid and liquid media was performed. The scanning electron microscope (SEM) investigated the morphological changes of treated fungi. 2, 6-Dichlorophenol Indophenol (DCPIP), drop collapse, emulsification activity, and oil Spreading assays investigated the biodegradation ability. The amount of produced biosurfactants was measured, and their safety profile was estimated by the germination assay of tomato seeds. Results: The tolerance test showed enhanced fungal growth of all isolates, whereas the highest dose inhibition response (DIR) was 77% for *Fusarium proliferatum* treated with the used oil (*p <* 0.05). SEM showed morphological changes in all isolates. DCPIP results showed that used oil had the highest biodegradation by *Fusarium verticillioides* and *Fusarium oxysporum.* Mixed oil induced the highest effect in oil spreading, drop collapse, and emulsification assay caused by *F. proliferatum*. The highest recovery of biosurfactants was obtained by the solvent extraction method for *F. verticillioides* (4.6 g/L), *F. proliferatum* (4.22 g/L), and *F. oxysporum* (3.73 g/L). The biosurfactants produced by the three isolates stimulated tomato seeds’ germination more than in control experiments. Conclusion: The current study suggested the possible oil-biodegradation activities induced by three *Fusarium* isolates from Riyadh, Saudi Arabia. The produced biosurfactants are not toxic against tomato seed germination, emphasizing their environmental sustainability. Further studies are required to investigate the mechanism of biodegradation activities and the chemical composition of the biosurfactants produced by these species.

## 1. Introduction

Oil hydrocarbons are organic compounds composed of carbons, nitrogen, and oxygen, which are the main structure of alkanes, cycloalkanes, and aromatic compounds [1]. Petroleum is produced by the thermal decay of buried organic matter over millions of years. Once extracted from the subsurface, crude oil (i.e., naturally occurring raw oil) is transferred to refineries where it undergoes distillation to produce a variety of products [2]. Depending on the proportions of constituents with heavy molecular weight, crude oil can be categorized as light, medium, or heavy oil [2]. Petroleum products are the most important sources of energy and are fundamental in industry, electricity production, transportation, and other essential processes. Petroleum pollution resulting from municipal and industrial activities might threaten the balance of the environmental ecosystem, and subsequently, the life of living organisms [1]. The extensive impact on the environment, specifically, the marine ecosystem is considered to be the main destination of petroleum hydrocarbon spills [3]. In order to recognize the purview of pollutant biodegradation, it is necessary to understand the properties of crude oil, mechanisms of natural petroleum bio-degradation, the fate of oil in the environment of concern, as well as factors that control the rate of biodegradation [2].

Crude oil (petroleum) is comprised of saturates/paraffin, resins, asphaltenes, and aromatics. Crude oil is an assortment of both simple and complex hydrocarbons that are degraded by a number of indigenous microorganisms, each capable of breaking down a specific group of molecules. Crude oil is a naturally occurring complex composed of hydrocarbon deposits and other organic materials. It is an important energy resource for daily life and in a number of different industries [4]. In different samples of crude oil, the same compound was degraded to different extents by the same organisms. This could be due to the bioavailability of a particular compound in a sample of crude oil, rather than its chemical structure. When at an appropriate concentration and composition, petroleum displays measurable toxicity toward living organisms. The toxicity varies widely and also depends on the organism’s biological state, species, and other environmental factors [5]. Approximately 85% of crude oil constituents are asphalt base, paraffin base, and/or mixed base [6]. Crude oil is categorized into four factions, Saturates (aliphatics), Aromatics (ringed hydrocarbons), Resins, and Asphaltenes. The composition of crude oil can vary depending on location, the relative age of an oil field, and the depth of the oil well. Used oil causes ground and surface water pollution, reduces oxygen supply to microorganisms, and accumulates toxic metal ions. Several chemical, physical, and biological approaches have been employed in an attempt to reduce the adverse effects of large-scale oil spillages, but most efforts have limitations in their applications. This is because they are either costly or put the ecosystem at risk [7]. 

Extensive efforts are carried out to determine the safest method to remove petroleum spills from contaminated environments [8]. Biodegradation of oil spills by microbial activity is one of the cheapest, safest, and most effective techniques used in oil remediation [9]. The biodegradation of hydrocarbon pollutants has been explored extensively in myriad research studies throughout the past decade [9,10,11,12]. Several studies indicated the crude oil biodegradation ability of some isolated fungi including *Aspergillus niger, Penicillium documbens, Cochliobolus lutanus,* and *Fusarium solani* [9,13]. Few studies conducted research on the biodegradation abilities of some isolated fungi from contaminated soils in Saudi Arabia, such as *Candida* spp. [14], *Cladosporium sphaerospermum, Eupenicillium hirayamae, Paecilomyces variotii* [15], *Aspergillus polyporicola, Aspergillus spelaeus, A. niger* [16], *Fusarium oxysporum,* and *Drechslera spicifera* [17].

The biodegradation ability of some fungal species extended beyond oil hydrocarbons to other industrial aromatic compounds. In the study conducted by Sale et al., (2019), *A. niger* causes the decolorization and degradation of some textile azo dyes, which altered the physio-chemical characteristics of textile dyes as confirmed by Gas chromatography-mass spectroscopy (GC-MS) [18]. A similar study showed that *Aspergillus flavus* and *Fusarium oxysporium* isolated from dye-contaminated soil had significant biodegradation capacities confirmed by GC-MS [19]. 

The current study aimed to investigate the biodegradation ability of *Fusarium* spp. isolated from contaminated soil samples in different crude oil reservoirs in Riyadh, Saudi Arabia. The study focused on three *Fusarium* isolates; *Fusarium verticillioides*, *Fusarium proliferatum*, and *F. oxysporum*, which to our knowledge, is the first biodegradation study of these species in Riyadh, Saudi Arabia. Furthermore, a review of the literature showed that the biodegradation ability of *F. proliferatum* was not screened before. Furthermore, the study aimed to show the possible fungal ultrastructural in the presence of crude oil as a result of mixing with crude oil and to show the difference between normal and treated species. Moreover, we screened the biosurfactant production and their effect on the characteristics of tested hydrocarbons, besides, screening the amount produced by each fungus and their effect on the germination of tomato seeds against water, in order to investigate their environmental tolerance and biosafety level. 

## 2. Materials and Methods

### 2.1. Soil and Hydrocarbons Collection

In the current study, the soil samples were collected from four different crude oil reservoirs. These fields were randomly selected in (JQGJ + G6) Al Faisaliyyah, (JQJ2 + P8) Al Sina’iyah, (JPMP + HX) Al Salhiyah, and (JP7M + PF) Ghubairah districts located in south and southeast of Riyadh city, Saudi Arabia. Briefly, at ~10 cm depth, 500 g of soil samples were collected in sterilized wide-mouth short-profile clear glass jars (Thermo Fisher Scientific Inc., Waltham, MS, USA) and stored at 4 °C until use. The collected samples were filtered to remove any soil debris using 2.5 mm pore-size sieves, and the color and pH of each were observed (Appendix A).

The hydrocarbon samples were collected from oil tankers of Aramco Company, Aramco District Cooling Plant, Dhahran heights road, 848F + 8W Dhahran, located the eastern province of Dammam, Saudi Arabia. The tested hydrocarbons included crude oil, used oil, diesel, and kerosene. Furthermore, by mixing equal volumes of each oil, another hydrocarbon was prepared and named (mixed oil). All samples were kept at 4 °C in sterile Erlenmeyer flasks with cap (Thermo Fisher Scientific Inc., Waltham, MS, USA). 

### 2.2. Isolation and Identification of the Fungal Strains

The fungal species in the collected soil samples were isolated as described before [20]. We used the dilution method, where 0.2 mL of each 10% soil sample (dissolved in distilled water) was added to a sterile potato dextrose agar (PDA) plate (Sigma-Aldrich Co., St. Louis, MO, USA). The plates were incubated at 30 °C for three days, where different fungal colonies started to grow (Appendix A). Each colony was isolated, re-cultured in a new PDA plate, and incubated at 30 °C for three days until pure cultures of fungi were obtained. The last step was repeated twice to increase the purity of the species. The pure cultures were transferred into McCartney bottles of PDA slant, incubated at 30 °C for three days, then stored at 4 °C [21]. 

The morphological identification of the Fusarium species was performed by microscopic investigation according to the morphological characteristics and taxonomic keys provided in the Pictorial atlas of Watanabe [22]. The mycological keys included the segmentation of the spores and conidiophores structures as described before [23]. In the current study, *F. verticillioides*, *F. proliferatum*, and *F. oxysporum* were identified.

### 2.3. Screening of the Biodegradation Ability of Fungi

#### 2.3.1. Growth of Fungi on Solid Media 

Mineral Salt Medium (MSM) agar was supplemented with 1% (*v*/*v*) of different hydrocarbons. 10 mL of prepared MSM/hydrocarbon mixture was introduced to each Petri dish, while another plate containing only 10 mL MSM was used as a control without any treatments. Each fungal strain was inoculated on separate Petri dishes to study its hydrocarbon tolerance capacity [21]. The mycelial growth was observed daily for ten days post-inoculation and the fungal dose inhibition response (DIR) to hydrocarbons was estimated at 1%, 5%, and 10% concentrations of each oil by calculating the percentage of the mycelial growth (cm) on the hydrocarbon plate to the control plate [16]. The experiment was repeated in triplicates.

#### 2.3.2. Growth of Fungi in Liquid Media

Erlenmeyer flasks of 200 mL MSM liquid culture medium mixed with 1% of one of the tested hydrocarbons were prepared as the sole carbon source for fungal growth. The filamentous fungi samples (*F. verticillioides*, *F. proliferatum*, and *F. oxysporum*) were inoculated by adding two small slices (0.5 × 0.5 cm) of the solid medium, previously prepared, to each Erlenmeyer flask. Flasks were incubated at 25 °C on an orbital shaker (140 rpm) for 30 days, while the growth rate was measured every three days by estimating the increase in the flask weight (gm) [24]. All experiments were carried out in triplicates.

The dry weight of each fungus in each treatment setting was estimated at the end of the 30 incubation days. The flask contents were filtered by Whatman^®^ qualitative filter paper, Grade 4 (Sigma-Aldrich Co.; St. Louis, MO, USA). The resulting mass was rinsed with 100 mL of chloroform and oven-dried at 60 °C overnight. The dry weight, measured by Kern analytical balance (0.1 mg–220 mg) (Sigma-Aldrich Co.; St. Louis, MO, USA) was calculated after cooling and compared to the biomass produced by the control. The experiment was repeated in triplicates.

#### 2.3.3. 2,6-Dichlorophenol Indophenol (DCPIP) Assay

DCPIP technique was employed to assess the oil-degrading ability of fungal isolates [25]. Briefly, Erlenmeyer flasks with 100 mL of MSM mixed with 1% of either one of the tested hydrocarbons, 0.1% Tween 80, and 0.6 mg/mL of redox indicator (DCPIP) (Sigma-Aldrich Co.; St. Louis, MO, USA) were autoclaved for 30 min. at 121 °C. Here, at one-week growth, 1 cm^2^ of fungal hyphae were inoculated and incubated on a shaker (140 rpm) for two weeks. The colorimetric changes of DCPIP were estimated, spectrophotometrically, at 420 nm by Multiskan SkyHigh Microplate Spectrophotometer (Thermo Fisher Scientific Inc., Waltham, MS, USA). The experiment was repeated in triplicates. 

### 2.4. Screening of the Fungal Ultrstructural Chnges by Scanning Electron Microscopy (SEM)

A large amount of the culture media from the previous experiment was used to investigate the induction of morphological changes in each of the tested species upon treatment with 1% of the crude petroleum oil. The experimental investigation was performed by the scanning electron microscope (SEM Quanta-250, FEI Company, Eindhoven, The Netherlands), as described before [16].

### 2.5. Screening for Biosurfactant Production

The cell-free metabolic liquid (CFS) containing biosurfactants was collected from the mycelial growth in the liquid MSM media, mentioned above. The media were centrifuged at 10,000 rpm and 4 °C for 30 min. The supernatants were collected and filter-sterilized through a sterile 0.45 μm pore size membrane. The following assays were used to estimate the amount and characteristics of the produced CFSs [26]. 

#### 2.5.1. Drop Collapse Assay

That assay measures the destabilization of liquid droplets by surfactants [26]. Briefly, 10 µL of CFSs were placed on glass slides containing 100 µL of crude oil, which was equilibrated for 1–2 h at room temperature. The picture was captured after 1 min using a light microscope. The spread of CFS at a diameter >0.5 mm was expressed by {+}, where the negative collapse was expressed as {−} to indicate the lack of biosurfactant production.

#### 2.5.2. Oil Spreading Assay

The oil spreading method is used to investigate the ability of the CFS to increase the oil displacement activity. Briefly, 20 µL of crude oil was carefully applied on the surface of a drop of water. The spread of the oil drop was noticed to investigate the formation of clear zones, which correlates to the biosurfactant activity. The plates were captured, and the diameter of the formed clear zones was compared to that of the negative control (water) and positive controls (Sodium dodecyl sulfate (SDS)) [16].

#### 2.5.3. Emulsification Activity

Emulsification activity correlates to the concentration of the biosurfactant, which was able to increase the oil emulsification Briefly, 2 mL of each of the tested hydrocarbons was added to a clean glass test tube, then 2 mL of CFS of one of the species was added to the top of the oil. Another tube was used as a control by replacing the CFS with water. The tubes were vigorously shaken by vortex to homogenize the contents for 2 min at room temperature. The mixtures were allowed to settle overnight, then the heights of the emulsion layers were measured and proportioned from the total height to estimate the percentage of the emulsification activity, as described before [27]. 

### 2.6. Recovery of Biosurfactants

To estimate the amount of biosurfactants produced by each fungal species by using different hydrocarbons as the sole carbon source. In the acid precipitation method, CFCs produced in each treatment were adjusted to acidic pH 2 by 6N HCl. After overnight incubation at 4 °C, the mixtures were mixed with equal volumes of chloroform and methanol (2:1), shaken, and re-incubated overnight at room temperature. Post-incubation a light brown color precipitated, which was further collected at 10,000 rpm, 4 °C for 30 min. The supernatant was discarded and the precipitate was dried and weighed [28]. 

In solvent extraction assay, CFSs were mixed with an equal volume of a solvent mixture of methanol, chloroform, and acetone (1:1:1). The mixtures were vigorously shaken for 5 h at 200 rpm and 30 °C. The mixture was separated into two layers, the lower white precipitate was separated, dried, and weighed [29].

Another two precipitation methods of biosurfactants ammonium sulfate and zinc sulfate were applied, as well [26]. CFSs were gently mixed with 40% (*w*/*v*) of either ammonium sulfate or zinc sulfate, incubated overnight at 4 °C, and spun down by centrifugation at 10,000 rpm for 30 min at 4 °C. In the ammonium sulfate method, the mixture was extracted with an equal volume of acetone, where a white creamy precipitate was formed, dried in a fume hood, and weighed. In the zinc sulfate method, light brown precipitates were formed directly, which were dried and weighed, accordingly. 

### 2.7. Germination Assay

To assess the sustainability of fungal biosurfactants, a germination assay was applied [27]. A hundred tomato seeds were immersed in 50 mL of CFS and incubated overnight on an orbital shaker at 200 rpm. Post-incubation, the seeds were rinsed with distilled water and seeded over pierced Whatman filter paper for 20 days with regular irrigation twice daily. The number of growing seeds CFS-treated (production of roots) was recorded daily and the seed germination rate was estimated by calculating the percentage of germinated CFS-treated seeds in the control setting (seeds were not treated with CFS). 

### 2.8. Statistical Analysis

The statistical analysis was performed by the IBM SPSS^®^ Statistics software, version 22, (Armonk, NY, USA). All of the above experiments were performed in triplicates. The Chi-square test was used and the statistical results were significant at *p <* 0.05.

## 3. Results and Discussion

In the current study, different soil samples were collected from four locations in Riyadh, Saudi Arabia. *Fusarium* species were detected in twelve soil samples from all investigational sites. The frequency of investigated fungi and the properties of soil samples are listed in Table 1. All locations for soil samples were in gas stations in Riyadh, where oil hydrocarbons were collected from the oil pipelines of Aramco company in Dammam. We selected the soil sample locations in Riyadh, as this location is far enough from the primary source of petroleum exploration sites and at the time it was one of the most crowded cities in Saudi Arabia, so we expect the availability of microbial species will be higher and that is reported in Table 1. In contrast, oil samples for our study were collected from the main pipelines to overcome any other source of microbial contamination.

Based on the physical and microscopic diagnosis, the soil containing the *Fusarium* fungal species was investigated (Appendix A). The microscopic investigation revealed the existence of three *Fusarium* species; *F. verticillioides, F. proliferatum,* and *F. oxysporum*. In the current study, the microbial identification of *Fusarium* spp. was carried out according to the microscopic characteristics of the segmentation of the spores and conidia/Conidiophores structures using the guidelines of the Pictorial atlas of soil and seed fungi of Watanabe [30]. A similar study used the same protocol in which the sporulated fungi were seeded on Potato Dextrose Agar (PDA) plate and only the microscopical identification was used [31]. This is common and reliable; besides, it was used as a single identification method that was published before. 

Initially, for *F. verticillioides*, cultures had white mycelia but may develop pink to violet pigments with age. The microconidia were small, hyaline, single-celled, and clustered in long chains and clusters. Microconidia were oval to club-shaped with a flattened base and 0-septate borne and composed of monophialides. Monophialides occurred in V-shaped pairs to give a rabbit ear appearance. Macroconidia were also observed but were few and difficult to find. Morphological characteristics include macroconidia which were relatively slender, sickle-shaped to almost straight, with 5–6 septate, a tapered apical cell, and a distinctly foot-shaped basal cell. 

Cultures of *F. proliferatum* had initially white abundant aerial mycelium. Violet pigments were produced in the media, but with overall pigmentation varying in intensity, from nearly colorless to almost black. The microconidia were club-shaped with a flattened base and clustered in chains and less commonly in false heads from monophialides and polyphialides. Macroconidia were slender, almost straight, and usually have 3–5 septate. Distinguishing characteristics of *F. proliferatum* from *F. verticillioides* were the extensive proliferation of the polyphialides and the shorter length of microconidia chains.

Colonies of *F. oxysporum* were growing rapidly, 4.5 cm in four days. The aerial mycelium was purple with discrete orange sporodochia present in some strains, and the hyaline reversed to dark blue or dark purple. Conidiophores were short, single, lateral monophialides in the aerial mycelium, and later arranged in densely branched clusters. Macroconidia were slightly curved, pointed at the tip, mostly three septate, basal cells pedicellate, and of 23–54 × 3–4.5 µm length. Microconidia were abundant, not-chained, mostly non-septated, ellipsoidal to cylindrical, straight or often curved, and 5–12 × 2.3–3.5 µm in length. Chlamydospores were terminal or intercalary, hyaline, smooth or rough-walled, and 5–13 µm in length. The phialides were short and mostly non-septated. That description agreed with morphological characteristics described before for *F. verticillioides* [9,31], *F. proliferatum* [5,24], and *F. oxysporum* [32].

In assessing the growth rates of the isolated fungal species on solid agar media, no clearing zones with abiotic control samples were found, which means that the clearance zones resulted from fungal action and not because of other abiotic factors (Table 2). Previous studies showed these clear zones were produced by an extracellular lipase enzyme, which catalysis hydrolysis reactions of the fats and maintain thermal flexibility around the site of microbial activity [33]. To test the hydrocarbon tolerance of each fungal species, the species were cultured on solid MSM media (Figure 1). In agreement with previous studies, this study found clearing zone formation after seven days with all the isolated fungi consortia.

The effect was estimated at three doses of each hydrocarbon and the ability to tolerate hydrocarbon presence was calculated based on DIR. Overall, *F. verticillioides* appeared to have the least tolerance for all the hydrocarbons tested in comparison to the other fungal species. The growth rate of *F. proliferatum* differed depending on the carbon source used. The best growth rate evidenced by larger clear halos around the colonies was observed in the dishes treated with used and crude oils, whereas the lowest was observed with diesel and mixed oils. In *F. oxysporum,* the highest effect was observed with crude oil and kerosene (Figure 2A). The DIR percentages were calculated for the lowest concentration (1%) of each oil (Figure 2B). The highest DIR% was for used oil 77.14% in *F. proliferatum* plates, whereas the lowest was for mixed oil by 42.22% with *F. verticillioides.*


Moreover, the hydrocarbon tolerance was tested in liquid MSM media supplied with 1% of different oils (Appendix A). The results showed that the growth rate of *Fusarium* differed depending on the carbon source used. That was evidenced by the change of color in each flask. The highest growth rate of *F. verticillioides* was observed with crude and used oil, whereas the lowest growth rates were observed with kerosene measured on the 30th day. For *F. proliferatum,* the highest growth rate was observed with crude oil and diesel, while the lowest was observed with used oil. Similarly, for *F. oxysporum* the highest growth rate was observed with crude oil, whereas the lowest was observed with diesel, used, and mixed oils (Figure 3). The comparison of the wet weight of all treatments on the 30th day (Figure 3D) showed that Crude oil produced the highest increment in the growth rate of 9.2 g for *F. oxysporum* and 8 g for *F. proliferatum*, where both crude and used oils had almost similar results by 7 and 7.3, respectively, for *F. verticillioides*. The lowest increments in the growth of *F. verticillioides* were induced by Kerosene (3.8 g) and mixed oil (3.9). For *F. proliferatum* and *F. oxysporum*, the lowest growth increment was induced by used oil (2.7 g and 3.4 g, respectively).

A comparison of the resulting dry weight represented the growth weight of *Fusarium* species on the 30th day (Figure 3, Table 3). By comparison to the control, all hydrocarbons increased significantly, and the growth of *F. verticillioides*, in which crude and used oils induced the highest and lowest dry weights, respectively (*p <* 0.05). For *F. proliferatum*, all hydrocarbons induced significantly higher growth than the control, except with the mixed oil (*p >* 0.05). Only crude oil, diesel, and kerosene had a significant dry weight of *F. oxysporum* on the 30th day. The results showed that fungi benefit from tested oils as sole carbon source. In conclusion, all *Fusarium* species preferred crude oil as the sole carbon source, whereas mixed and used oils were less effective, which might be due to higher impurities and different chemical content.

Based on the result of this study and previous studies, the presence of clearing zones in oil plates is evidence of oil biodegradation by the fungal cells themselves and also by their extracellular metabolic products. In agreement with our results, a previous study revealed that the fungal load of *Fusarium* spp. increased up to the sixth week with 10% of crude oil (1.41 × 10^5^ ± 0.06 cfu/g) [34]. Other studies revealed the tolerance of kerosene and diesel as the sole carbon source for the growth of *Fusarium* spp. [15,35]. 

A similar study was conducted by Marchand et al., (2017), where the hydrocarbon-degrading capabilities of bacteria were documented via spectrophotometry and were analyzed at 400 nm wavelength, whereas the degradation kinetics were studied at 540 nm [36]. In another study, *A. niger* and *P. documbens* were identified as very efficient fungal isolates, showing high biodegradable activity [13]. *A. niger* was shown to actively degrade four types of hydrocarbons [37]. Additionally, *Aspergillus* spp.; *Penicillium* spp.; *Helminthosporium* spp.; *Rhizopus* spp.; and *Fusarium* spp. were reported to be efficient metabolizers of hydrocarbons [38]. Fungi associated with the *Cunninghamella*, *Penicillium*, *Fusarium*, *Aspergillus,* and *Mucor* species are generally involved with the more obstinate degradation of aromatic hydrocarbons [39]. Microbial hydrocarbon degradation involves complex enzymatic activities such as those of hydroxylases, dehydrogenases, monooxygenases, dioxygenases, oxidoreductases, etc. [21]. Fungi facilitates the degradation of recalcitrant hydrocarbons by secreting extracellular enzymes that transform the hydrocarbons into intermediates with lower toxicity, which then leads to further degradation by bacteria [40]. 

The higher growth rate might be due to several morphological changes that occurred upon treatment with hydrocarbons. In this regard, the current study investigated the morphological characteristics of isolated *Fusarium* species by SEM imaging. As shown in Figure 4, SEM images of the untreated *F. verticillioides* showed healthy, slender, and uniform mycelium with a smooth surface and intact structures. The treatment with 1% crude oil caused swellings and blurring edges at the knots, where spores had irregular surfaces and edges, which is in contrast to the control.

Although *F. proliferatum* showed an ability to grow on 1% crude oil, that caused some deformities in its cellular growth. Similar results were obtained for *F. proliferatum* (Figure 5). The SEM images revealed swelling, blurring, and knots forming on the outmost edges, which decreased the smoothing of the cellular walls. The microconidia structures were not clear or disappeared. 

In the case of *F. Oxysporum,* the untreated fungus had lean, smooth, and uniform mycelia, whereas the outmost surfaces had intact structures. In the crude oil-treated images, the mycelia had abnormal swelling, proximal knots, and irregular edges. The microconidia or spores had unhealthy morphology with segmented uneven edges and irregular surfaces (Figure 6). 

So, the current study showed abnormal morphology of the *Fusarium* spp. in the crude oil treatment, despite the improved growth rate. A previous study reported similar effects of some essential oils on the spores of *F. oxysporum*, which lost their cytoplasmic content resulting from swelling and severe rapture of the cellular wall [41]. Another study showed that *F. oxysporum* f. sp. cubense treated with antifungal hydrocarbons had wrinkled external hyphal surfaces, borders deformity, and ruptured bubbles [42]. Another study showed that the filamentous fungi *D. spicifera* showed abnormal morphological changes upon treatments with different petroleum hydrocarbons [27]. 

The ability of isolated species as oil-degraders was tested by DCPIP assay (Appendix A). The clear decolorization of DCPIP dye indicated the biodegradation activity of *Fusarium* species extracts against different hydrocarbons. As shown in Figure 7, the O.D. results demonstrated that all isolates exhibited a biodegrading ability against all hydrocarbons. *F. verticillioides* exhibited the highest O.D. on the 15th day with used oil (0.557) and the lowest with crude oil (0.312) and kerosene (0.319). In contrast, for *F. proliferatum*, the highest color change appeared with crude oil (0.459), whereas the lowest was with diesel (0.365). Alternatively, *F. oxysporum* exhibited the highest biodegrading ability against used oil (0.436), whereas the lowest was for diesel (0.336). 

In agreement with our results, a previous study reported the biodegradation of engine oil by 89% when incubated with *Fusarium* spp. isolated from Western India [43]. Another study showed the biodegradation ability of *F. solani* a week post-incubation with 1% crude oil as indicated by the colorless of DCPIP [44]. Another previous study used two *Penicillium* spp. strains and showed their ability to degrade crude oil by 57% [45]. All these studies confirm the biodegradation ability of fungal and bacterial strains isolated from contaminated soils against different hydrocarbons.

A major limiting factor in hydrocarbon remediation is the low bioavailability of hydrocarbon components [46]. In this case, biosurfactants interact with phase boundaries in a heterogenous system to solubilize organic compounds, acting as surface-active amphiphilic compounds with a hydrophobic and hydrophilic moiety [47]. This circumstance improves bioavailability by enhancing hydrocarbon solubilization in water, or vice versa [48]. Biosurfactants play significant roles in a variety of fields, including biodegradation, biodegradation, and oil recovery. Compared to chemical surfactants (sulfonates, sulfates, carboxylates) [9], biosurfactants have much lower toxicity and higher biodegradability. 

In the current study, the drop-collapse assay was used to assess the effect of the biosurfactants produced by *Fusarium* spp. on the physical properties of tested hydrocarbons (Table 4). Compared to positive and negative controls, the highest drop-collapse was induced by mixed oil, followed by crude and used oil. A weaker collapsing effect was observed in diesel and kerosene.

Another effect, the oil-spreading activity, was tested with CFS of *F. verticillioides* against all tested hydrocarbons (Figure 8). The results showed that CFS had the highest effect on the spreading of crude oil. Another characteristic, the emulsification activity induced by CFSs of different *Fusarium* isolates was tested, as well (Appendix A). The mixtures in all tubes were separated into two layers, which indicates the presence of biosurfactants with varying levels. The highest emulsification was for mixed oil (63%) treated with the CFS of *F. proliferatum,* whereas the lowest was for crude oil (41.17%) treated with *F. oxysporum* (Table 5). 

A review report showed that that biosurfactants produced by *Pseudomonas* sp., *Bacillus* sp., *Citrobacter freundii*, and *Candida tropicalis* were able to biodegrade the heavy metals in the contaminated soil samples isolated from contaminated environments [49]. Furthermore, these biosurfactants are glycoconjugates, which assist in the reduction in surface tension and increasing of the solvent interface, and that plays critical role in increasing the bioavailability of organic pollutants, such as oil-hydrocarbons [49].

Although most of the biosurfactants are of bacterial origin with producers affiliated to the *Pseudomonas*, *Acinetobacter*, and *Bacillus* genera [50], the production of biosurfactants by yeasts and filamentous fungi has been further scrutinized in recent years. Fungal biosurfactant producers are affiliated with the *Candida*, *Pseudozyma*, or *Rhodotorula* genera for yeasts and the *Cunninghamella*, *Fusarium*, *Phoma*, *Cladophialophora*, *Exophiala*, *Aspergillus*, and *Penicillium* genera for filamentous fungi [51]. A recent study underscored that some microbial strains could lead to improved hydrocarbon degradation by utilizing crude oil components as carbon sources in order to synthesize biosurfactants [52]. 

Several recovery assays were performed to measure the dry weight of the biosurfactants produced by each species treated with different hydrocarbon sources (Table 6). 

As shown in the previous table, amounts of biosurfactants differed due to the type of species, hydrocarbon, or even the recovery assay used. Overall, both ammonium and zinc sulfate precipitation methods produced the lowest amounts of biosurfactants. The solvent extraction method was the best producer of surfactants in all hydrocarbon treatments, except for mixed oil as acid precipitation was the best choice. For used oil and diesel, the highest amounts were produced by *F. oxysporum*, whereas for crude oil and kerosene, *F. verticillioides* was the highest biosurfactant producer. *F. proliferatum* produced the highest amount of biosurfactants by mixed oil treatment. All these findings suggested that different compositions of hydrocarbons might be preferred by specific *Fusarium* species as the sole carbon source, which might reflect its biodegradation capacity. 

In agreement with the current study, a previous study showed that *Fusarium* sp. BS-8 with a 71% hydrocarbon emulsifying index induced the recovery of 5.25 g/L of crude Lipopeptide biosurfactant. Another study showed that *F. oxysporum* produced 1.02 g/L of fatty acid biosurfactant when treated with crude oil [53]. Similarly, another study showed the ability of *Fusarium fujikuroi*, isolated from contaminated soil of an oil reservoir in Brazil, to produce α, β-trehalose surfactant [54]. Moreover, an isolated strain of *F. proliferatum* from contaminated rice-bran oil sludge was able to produce an emulsifying biosurfactant against kerosene, n-dodecane, and refined oil. All these studies in compliance with our results highlighted the biosurfactants recovery capacity of *Fusarium* species. 

One of the aims of that research was to assess the impact of biopreparations containing biosurfactants and microorganisms on the germination and growth of plants predicted by the revitalization of soil contaminated by oil. The study showed that stimulated tomato seeds with biosurfactants germinated better than in control experiments (Figure 9). In this experiment, we showed the effect of isolated biosurfactants on the growth of tomato seeds, which is clear in Figure 9 at the 20th day post-treatment, as we can identify the roots of some seeds began to grow. The number of germinated seeds was mostly higher by 4–7% in comparison with seeds germinated in control plants used in this experiment (Figure 9B). The best germination was induced by CFS from *F. proliferatum* at the 20th day, while *F. verticillioides,* and *F. oxysporum* had a sudden decrease in germination level at the 15th and 18th days, respectively. That might be due to the different composition of these biosurfactants, as the content of carbon source might be varied among different CFS’s produced by different species. In addition, some of the growing roots were lost, even in control settings, due to the effect of dryness caused by aeration.

The biosafety levels of fungal recovered biosurfactants were reported in many studies. In the study conducted by Qazi and colleagues, 2013, the brine shrimp assay was used to assess the cytotoxic effect of biosurfactant produced by *Fusarium* sp. BS-8, which revealed high inactivity that did not affect the growth of shrimp eggs [54]. Similarly, the germination assay of cabbage seeds treated with biosurfactant produced by *Serratia marcescens* UCP 1549 revealed the highest biosafety levels of microbial biosurfactants [55]. The current findings and the previous studies indicated the non-inhibitory effects of biosurfactants, which confirm their eco-friendly characteristics and lower cytotoxically impact.

## 4. Conclusions

The current study highlighted the biodegradation capacities of three *Fusarium* species, isolated from contaminated soil samples, against different hydrocarbons obtained from oil reservoirs in Saudi Arabia. All isolates had clear ultrastructural modifications when incubated with crude oil compared to the control. *Fusarium* spp. were able to produce sufficient amounts of biosurfactants, which were able to increase the emulsification activity of tested oils. CFSs produced by *F. proliferatum* were more efficient than *F. verticillioides* and *F. oxysporum* in keeping longer germination of the tomato seeds, which suggests further studies to investigate the chemical composition and carbon content of these biosurfactants. The study also suggested isolated fungi as a promising microbial resource for the biodegradation of crude oil pollution. Some limitations occurred regarding the specific chemical composition of crude and used oils, as we consider its effect as a natural carbon-source, unlike other hydrocarbons of known components such as kerosene and diesel oils. That raised the need for further analyses to accurately determine the composition of these biosurfactants and the degradation dynamics.

## Figures and Tables

**Figure 1 jof-09-00216-f001:**
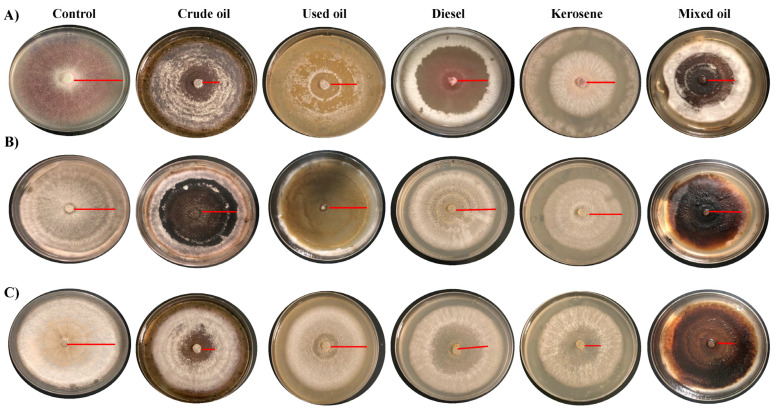
Fungal growth in solid MSM media with different carbon sources. Isolated species were cultured in Petri dishes containing Mineral Salt Medium (MSM) agar supplemented with 1% (*v*/*v*) of different hydrocarbons and incubated for a week. Red lines indicated the growth area of each species from borders to the dish center. (**A**) *F. verticillioides*, (**B**) *F. proliferatum* (**C**) *F. oxysporum*.

**Figure 2 jof-09-00216-f002:**
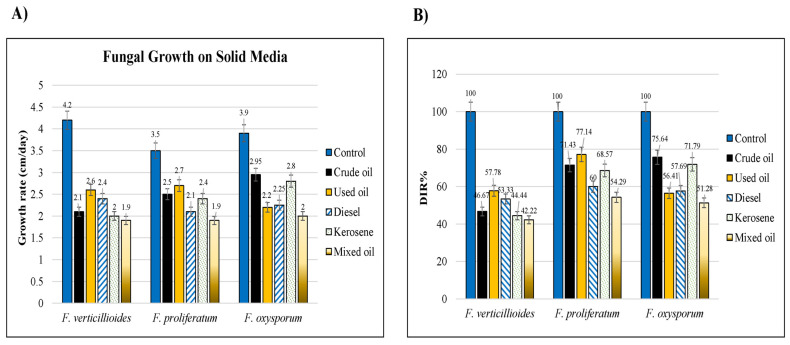
Charts showed the hydrocarbons tolerance of three *Fusarium* species growing on MSM solid medium with 1% of the studied hydrocarbons versus the control (water). (**A**) growth rate expressed in cm/day, (**B**) DIR percentages at 1% concentrations.

**Figure 3 jof-09-00216-f003:**
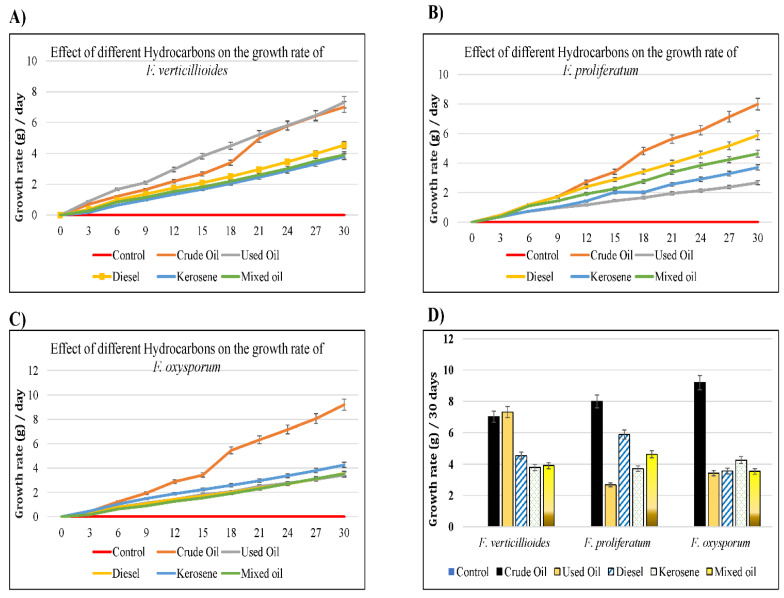
Charts showed the hydrocarbons tolerance of three *Fusarium* species growing on MSM liquid medium with 1% of the studied hydrocarbons versus the control (water) for 30 days. (**A**) growth rate of *F. verticillioides*; (**B**) growth rate of *F. proliferatum*, (**C**) growth rate of *F. oxysporum,* (**D**) comparison of the net increment of the growth rates of isolated species on the 30th day.

**Figure 4 jof-09-00216-f004:**
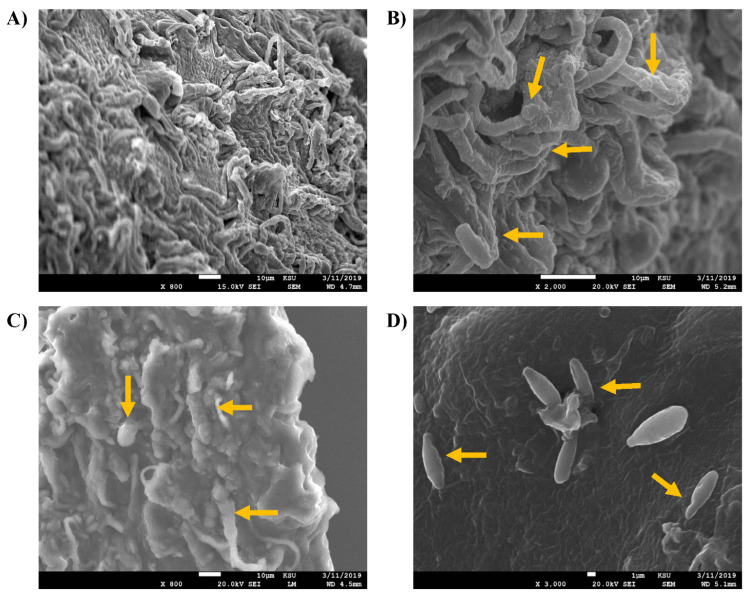
SEM images of *F. verticillioides.* The tested slides were loaded with *F. verticillioides* upon treatment with 1% of crude oil. (**A**) (At 800×, scale bar 10 µm) normal morphology of the healthy control, (**B**) (At 2000×, scale bar 10 µm), (**C**) (At 800×, scale bar 10 µm), and (**D**) (At 3000×, scale bar 1 µm) showed abnormal configuration of the fungus treated with 1% crude oil, indicated by the yellow arrows, (**B**,**C**) showing the swelling of edges, (**D**) showed the deformed spores.

**Figure 5 jof-09-00216-f005:**
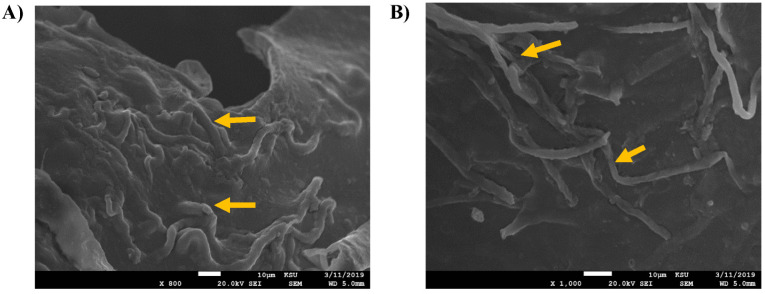
SEM images of *F. proliferatum.* The tested slides were loaded with *F. proliferatum* upon treatment with 1% crude oil. (**A**) (At 800×, scale bar 10 µm) and (**B**) (At 1000×, scale bar 10 µm) normal morphology of the healthy control, (**C**) (At 2000×, scale bar 10 µm), and (**D**) (At 1000×, scale bar 10 µm) showed abnormal configuration of the fungus treated with 1% crude oil, indicated by the yellow arrows, (**C**,**D**) showing the swelling of edges.

**Figure 6 jof-09-00216-f006:**
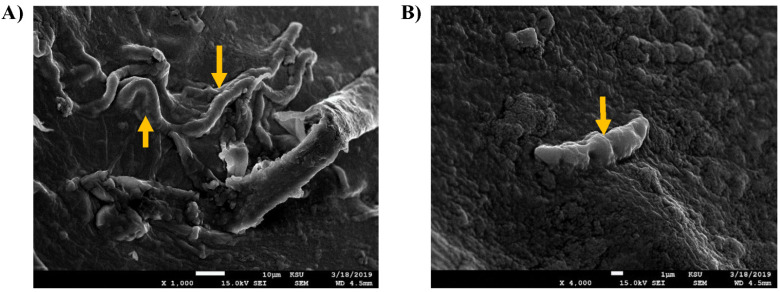
SEM images of *F. oxysporum*. The tested slides were loaded with *F. oxysporum* upon treatment with 1% of crude oil. (**A**) (At 1000×, scale bar 10 µm) and (**B**) (At 4000×, scale bar 1 µm) are the images of the untreated control sample. (**C**) (At 2000×, scale bar 10 µm) and (**D**) (At 5000×, scale bar 1 µm) are the images of fungus treated with 1% crude oil. The yellow arrows in (**A**,**C**) showed the mycelia of the control and treated fungus, respectively. (**B**,**D**) showed the spores morphology in the control and crude oil-treated settings.

**Figure 7 jof-09-00216-f007:**
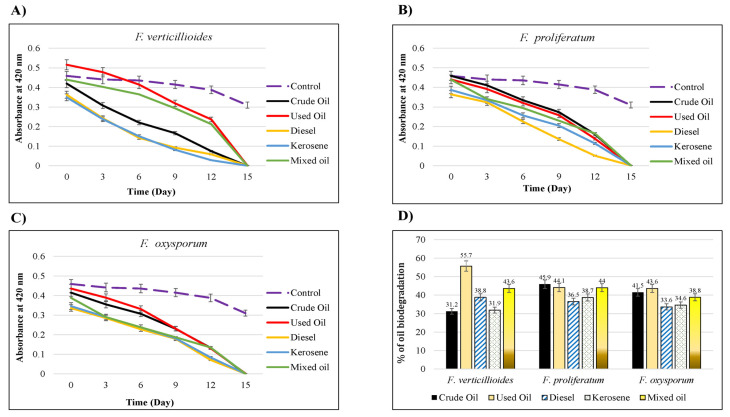
The oil biodegrading ability of tested isolates against different carbon sources by DCPIP assay. DCPIP technique was employed to assess the oil-degrading ability of fungal isolates [36,37,38]. MSM media supplemented with 1% of each hydrocarbon was mixed with 0.1% Tween 80 and 0.6 mg/mL DCPIP, then 1 cm^2^ of the fungal hyphae were added and incubated for two weeks. DCPIP results was obtained by measuring the colorimetric change, at 420 nm of different hydrocarbons with or without the fungal species for 15 days. (**A**) *F. verticillioides*, (**B**) *F. proliferatum*, (**C**) *F. oxysporum*, (**D**) Comparison of oil biodegradation percentage of all isolates at the 15th day post-incubation.

**Figure 8 jof-09-00216-f008:**
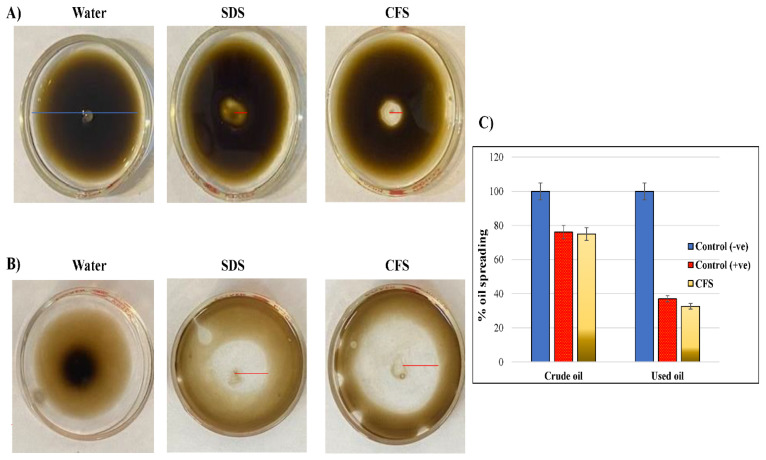
Oil spreading assay. 20 mL of water was added to the petri plate (≈5 cm × ≈5 cm) followed by 20 µL of crude oil making a thin layer on the surface of the water. A 10 µL aliquot of supernatant (CFs) was delivered onto the surface of the oil. (**A**) Crude oil (**B**) Mixed oil, (**C**) Bar chart to compare the effect of CFS on the spreading of different oils.

**Figure 9 jof-09-00216-f009:**
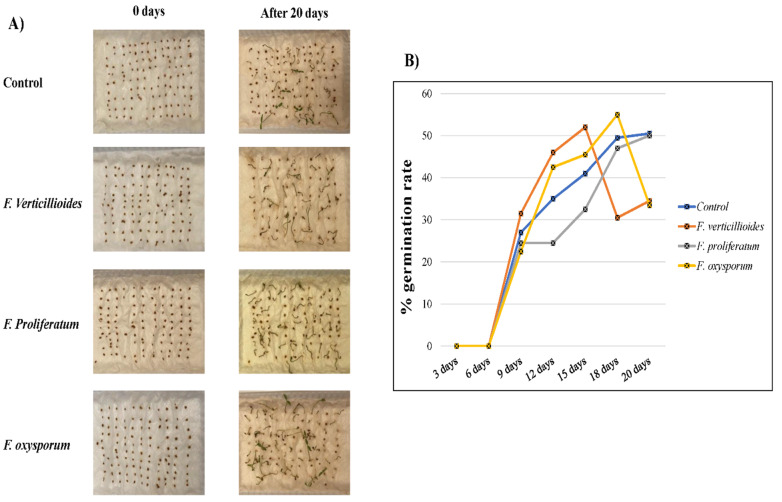
Assessment of toxicity the CFS containing biosurfactants produced by the *Fusarium* isolates on tomato seeds germination. (**A**) Germination assay of 100 toto seeds in either control settings (water) or treatment (CFSs of *Fusarium* isolates produced by 1% crude oil, (**B**) percentage of germination rate for 20 days post-treatment.

**Table 1 jof-09-00216-t001:** Frequency of isolated and identified fungi in each soil sample.

Characteristics of Different Soil Samples								
Location	Al Faisaliyyah		Al Sina’iyah		Al Salhiyah		Ghubairah	
pH	5.89		7.72		8.12		7.64	
Soil Color	Umber-brown		Caramel-brown		Ochre-yellow		Mocha-brown	
Incidence of microbes in each soil sample								
Strains	N	%	N	%	N	%	N	%
*Alternaria* spp.	4	26.67	2	13.33	0	0	2	13.33
*Aspergillus* spp.	6	40.00	2	13.33	4	26.67	2	13.33
*Drechslera* spp.	4	26.67	0	0	0	0	2	13.33
*Fusarium* spp.	4	26.67	4	26.67	2	13.33	2	13.33
*Mucor* spp.	4	26.67	2	13.33	2	13.33	0	0
*Penicillium* spp.	4	26.67	6	40.00	2	13.33	4	26.67
*Trichoderma* spp.	8	0	0	0	4	26.67	4	26.67
*Rhizopus* spp.	4	26.67	0	0	0	0	2	13.33
*Candida* spp.	2	13.33	2	13.3	0	0	2	13.33

**Table 2 jof-09-00216-t002:** The ability of isolated fungal strains to grow and tolerate different types of petroleum hydrocarbons.

Growth Rate		Control	Crude Oil	Used Oil	Diesel	Kerosene	Mixed Oil
*F. verticillioides*	1%	4.50	2.10	2.60	2.40	2.00	1.90
	5%	4.50	1.05	1.38	1.34	1.16	1.05
	10%	4.50	0.63	0.70	0.79	0.60	0.74
	DIR (%)	100.00	46.67	57.78	53.33	44.44	42.22
	*p*-value		<0.001 *	<0.001 *	<0.001 *	<0.001 *	<0.001 *
*F. proliferatum*	1%	3.50	2.50	2.70	2.10	2.40	1.90
	5%	3.50	1.43	1.59	1.01	1.15	0.95
	10%	3.50	0.70	0.92	0.80	0.96	0.70
	DIR (%)	100.00	71.43	77.14	60.00	68.57	54.29
	*p*-value		0.004 *	0.02 *	<0.001 *	0.002 *	<0.001 *
*F. Oxysporum*	1%	3.90	2.95	2.20	2.25	2.80	2.00
	5%	3.90	1.77	1.19	1.26	1.48	1.16
	10%	3.90	0.91	0.77	0.68	1.06	0.56
	DIR (%)	100.00	75.64	56.41	57.69	71.79	51.28
	*p*-value		0.01 *	<0.001 *	<0.001 *	0.005 *	<0.001 *

* Significant *p*-value < 0.05.

**Table 3 jof-09-00216-t003:** Biomass production of filamentous fungi with different carbon sources.

Growth Rate		Control	Crude Oil	Used Oil	Diesel	Kerosene	Mixed Oil
*F. verticillioides*	Weight (g)	0.67 ± 0.03	3.32 ± 0.03	2.4 ± 0.01	2.7 ± 0.04	2.4 ± 0.05	3.1 ± 0.15
	*p*-value		0.001 *	0.035 *	0.011 *	0.031 *	0.003 *
*F. proliferatum*	Weight (g)	1	3.3 ± 0.03	3.4 ± 0.25	3.3 ± 0.25	3.2 ± 0.01	1.3 ± 0.05
	*p*-value		0.02 *	0.015 *	0.023 *	0.029 *	0.803
*F. Oxysporum*	Weight (g)	1.3 ± 0.05	4.9 ± 0.033	2.8 ± 0.05	3.8 ± 0.06	4.6 ± 0.07	1.8 ± 0.03
	*p*-value		<0.001 *	0.154	0.023 *	0.002 *	0.642

* Significant *p*-value < 0.05.

**Table 4 jof-09-00216-t004:** Drop collapse test of CFSs produced by different strain of *Fusarium* spp.

Microorganism	Hydrocarbon Source						
	−C	+C	Crude Oil	Used Oil	Diesel	Kerosene	Mixed Oil
*F. verticillioides*	–	+++	++	++	+	+	+++
*F. proliferatum*	–	+++	++	++	+	+	+++
*F. oxysporum*	–	+++	++	++	+	+	++

−C: negative control (culture broth), +C: positive control (Triton X-100), –: negative collapse, +, ++, +++: degrees of positive collapse.

**Table 5 jof-09-00216-t005:** Emulsification activity of CFSs (%) produced by different strain of *Fusarium* spp.

Microorganism	Crude Oil	Used Oil	Diesel	Kerosene	Mixed Oil
*F. verticillioides*	47.05	59.09	60	54.54	56
*F. proliferatum*	52.94	52.38	52.38	57.14	63
*F. oxysporum*	41.17	57.14	54.54	50	56

**Table 6 jof-09-00216-t006:** Recovery of biosurfactants produced by *Fusarium* spp. (g/L).

Recovery Assay		Crude Oil	Used Oil	Diesel	Kerosene	Mixed Oil
Acid precipitation	*F. verticillioides*	3.2	3.53	1.3	1.7	2.3
	*F. proliferatum*	3.6	3.21	1.8	1.39	2.12
	*F. oxysporum*	2.37	4.31	1.7	1.5	2.01
Solvent extraction	*F. verticillioides*	4.6	4.39	2.32	2.81	1.93
	*F. proliferatum*	4.22	3.7	2.28	2.04	1.75
	*F. oxysporum*	3.73	5.35	2.35	2.35	1.51
Ammonium sulfate precipitation	*F. verticillioides*	0.03	0.05	0.076	0.118	0.089
	*F. proliferatum*	0.03	0.07	0.106	0.119	0.089
	*F. oxysporum*	0.03	0.11	0.102	0.084	0.05
Zinc sulfate precipitation	*F. verticillioides*	0.088	0.12	0.123	0.141	0.089
	*F. proliferatum*	0.088	0.12	0.11	0.141	0.07
	*F. oxysporum*	0.119	0.12	0.154	0.131	0.12

## Data Availability

All data are available in the article.

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
