# Peer review of "Biodegradation of Selected Hydrocarbons by Fusarium Species Isolated from Contaminated Soil Samples in Riyadh, Saudi Arabia"

_jof, 2023, doi:10.3390/jof9020216_

Round 1

Reviewer 1 Report (Previous Reviewer 1)

The manuscript have been improved in the revised file. However, I still recommend the authors consider the methodology used in the manuscript because it may confuse to the readers. The name of microorganisms should be written in abbreviation format after the first description.

Author Response

Reviewer No.1

Comment No:1: The manuscript has been improved in the revised file. However, I still recommend the authors consider the methodology used in the manuscript because it may confuse the readers. The name of microorganisms should be written in abbreviation format after the first description.

Response: Thanks for the reviewer's comment. We agree partially with his/her suggestion. We have made some modifications in the methodology section to clarify that point. Also, we have checked all the microorganisms’ names for the required comment.

Reviewer 2 Report (Previous Reviewer 2)

Paper entitled “Biodegradation of selected hydrocarbons by Fusarium species isolated from contaminated soil samples in Riyadh, Saudi Arabia” describe the efficacy of three Fusarium species isolated from contaminated sites to degrade the different hydrocarbons. The authors used SEM to investigate the morphological changes of treated fungi. Also, the biodegradation efficacy was checked by 2, 6-Dichlorophenol Indophenol (DCPIP), drop collapse, emulsification activity, and oil Spreading assays. Although the manuscript was improved, it needs other changes to be acceptable for publication in JOF. 

1-    The scientific names must be italics, for instance, line 91,

2-    Lines 106 – 108, Please add the coordinate for the collected oil samples used for fungal isolation.

3-    Line 115, “(Dammam, Saudi Arabia)” adding coordinates.

4-    For fungal identification, please add the culture and reverse photo and microscopic examination of three Fusarium isolates

5-    What about the optimization study such as temperature, pH another carbon source, nitrogen source, and inoculum size,… these parameters have critical roles in biodegradation

6-    The seed germination figure 9 is not clear, authors can select one or two seedlings to photo.

7-    The germination rate due to F. oxysporum and F. verticillioides decreased after 18 days, who? Please clarify.

8-    The conclusion should be rephrased to show the advantages and limitations of the current study.

9-    Data in Tables does not undergo statistical analysis.

Author Response

Reviewer No.2

Comment No 1: The scientific names must be italics, for instance, line 91,

Response: Thanks for the reviewer's comment. We agree with his/her suggestion. We have made the requested modifications.

Comment No 2: Lines 106 – 108, Please add the coordinate for the collected oil samples used for fungal isolation.

Response: Thanks for the reviewer's comment. We agree with his/her suggestion. We have made the requested modifications.

Comment No 3: Line 115, “(Dammam, Saudi Arabia)” adding coordinates.

Response: Thanks for the reviewer's comment. We agree with his/her suggestion. We have made the requested modifications.

Comment No 4: For fungal identification, please add the culture and reverse photo and microscopic examination of three Fusarium isolates

Response: Thanks for the reviewer's comment. Unfortunately, we don’t have this information, however, deep microscopic analysis of the organisms is shown in the SEM images. Also, the supplementary figure S2, showed culture photos of these organisms, captured by common camera.

Comment No 5: What about the optimization study such as temperature, pH another carbon source, nitrogen source, and inoculum size,… these parameters have critical roles in biodegradation

Response: Thanks for the reviewer's comment. We already added the pH degree of the soil samples in the results section. For the other variables, we tried to keep all the physical conditions such as temperature, nitrogen source, and inoculum size constant at all experimental steps. That helped us to focus on the biodegradation ability of these hydrocarbons without caring for the critical roles of other parameters.  Also, many similar studies follow the same experimental design. some of them are mentioned here:

  1. Ozyurek, S. B.; Bilkay, I. S. Comparison of petroleum biodegradation efficiencies of three different bacterial consortia determined in petroleum-contaminated waste Mud Pit. SN Appl. Sci. 2020, 2, 272. https://doi.org/10.1007/s42452-020-2044-5.
  2. Olukunle, O. F.; Oyegoke, T. S. Biodegradation of Crude-oil by Fungi Isolated from Cow Dung contaminated Soils. Niger. J. Biotechnol. 2016, 31, 46-58. https://doi.org/10.4314/njb.v31i1.7.
  3. Sajna, K. V.; Sukumaran, R. K.; Gottumukkala, L. D.; Pandey, A. Crude oil biodegradation aided by biosurfactants from Pseudozyma sp. NII 08165 or its culture broth. Bioresour. Technol. 2015, 191, 133–139.
  4. Marchand, C.; St-Arnaud, M.; Hogland, W.; Bell, T. H.; Hijri, M. Petroleum biodegradation capacity of bacteria and fungi isolated from petroleum-contaminated soil. Int. Biodeterior. Biodegradation 2017, 116, 48–57. https://doi.org/10.1016/j.ibiod.2016.09.030.

Comment No 6: The seed germination figure 9 is not clear, authors can select one or two seedlings to photo.

Response: Thanks for the reviewer's comment. However, we don’t find any necessities for doing that. In this experiment, we tried to show only the effect of biosurfactants on the growth of tomato seeds, which is clear in figure 9 at the 20th day post-treatment, as we can identify the roots of some seeds began to grow. Few modifications were made in the results section to clarify this point.

Comment No 7: The germination rate due to F. oxysporum and F. verticillioides decreased after 18 days, who? Please clarify.

Response: Thanks for the reviewer's comment. We agree with his/her suggestion. That might be due to different composition of these biosurfactants, as the content of carbon source might be varied among different CFS’s produced by different species. Few modifications were made in the results section to clarify this point.

Comment No 8: The conclusion should be rephrased to show the advantages and limitations of the current study.

Response: Thanks for the reviewer's comment. We agree with his/her suggestion. We have made the requested modifications.

Comment No 9: Data in Tables does not undergo statistical analysis.

Response: Thanks for the reviewer's comment. We don’t agree with his/her suggestion. The statistical analysis is added in tables 2 and 3. These tables compare different parameters of the hydrocarbon treatments against control. The other table is not scientifically amended to have control results (or it will be simply equal zero) so no statistical analysis can be applied here.   

Reviewer 3 Report (New Reviewer)

Review Report of Manuscript No. jof-2161801    

The manuscript entitled “Biodegradation of selected hydrocarbons by Fusarium species isolated from contaminated soil samples in Saudi Arabia is quite interesting. The topic selection of the manuscript is good, in line with the current hot spots in the environmental field. In totality, this paper is pleasant to read, well-structured and well-written. However, it needs some corrections and there are some queries which the authors should kindly respond to make it good.

Some specific suggestions or questions are listed below:

1. The Abstract should be written more precisely and explain novelty of this work.

2. Introduction is easy to read but needs a little completed. I suggest this section can be shortened. Introduction should briefly place the study in a broad context and highlight why it is important. It should define the purpose of the work and its significance. The current state of the research field should be reviewed, and key publications cited. Finally, briefly mention the main aim of the work and highlight the principal conclusions.

3. Line 78-83: Please check all the species names. Species names are typically given in full the first time they are used within the main text and then abbreviated throughout the remainder of the text.

4. Line 167: Provide the longitude and latitude here.

5. Line 177: Screening for Biosurfactant Production, please include related references to support the method.

6. Results and discussion, where are the identification results? Please add detailed identification results here.   

7. Line 242: Table 1, use the three-dimensional diagram for the table. The same as other tables.

8. Line 363: Table 3, some parts are missing in the table.

9. Line 371: Leahy et al. (2003), please check the format.

10. Line 378-383: Fusarium strains are known for their metabolic capability to degrade various organic pollutants such as Biodegradation of allethrin by a novel fungus Fusarium proliferatum strain CF2, isolated from contaminated soils. Microorganisms, 2020. Why the authorS did not compare the degradation activity of F. verticillioides and F. proliferatum with that of other Fusarium strains based on the literature. Authors should add more information into this section and cite the recent research into the field.

11. Line 564-609: Authors should add more information into this section and cite the recent research into the field such as: Biosurfactant is a powerful tool for the bioremediation of heavy metals from contaminated soils. Journal of Hazardous Materials, 2021; Microbial glycoconjugates in organic pollutant bioremediation: recent advances and applications. Microbial Cell Factories, 2021. This way the authors will demonstrate that they really have a good knowledge of the related literature.

12. Conclusion: This section should be revised for the better understanding of the topic and its future research.

13. References: Many of the references have been superceded and more modern ones are required such as Microbiol. Rev. 1990, 54, 305–315; Biotechnol. Tech. 1993, 7, 745–748; Int. Biodeterior. Biodegradation. 2004, 54(1), 61-67.

Author Response

Reviewer No.3

Comment No 1: The Abstract should be written more precisely and explain novelty of this work.

Response: Thanks for the reviewer's comment. We agree with his/her suggestion. We have made the requested modifications.

Comment No 2: Introduction is easy to read but needs a little completed. I suggest this section can be shortened. Introduction should briefly place the study in a broad context and highlight why it is important. It should define the purpose of the work and its significance. The current state of the research field should be reviewed, and key publications cited. Finally, briefly mention the main aim of the work and highlight the principal conclusions.

Response: Thanks for the reviewer's comment. We agree with his/her suggestion. However, it was requested by other reviewers, from the previous version to expand the introduction section. Besides, the current version of introduction highlighted the importance of the study, aims, and key publications as required by the reviewer.

Comment No 3: Line 78-83: Please check all the species names. Species names are typically given in full the first time they are used within the main text and then abbreviated throughout the remainder of the text.

Response: Thanks for the reviewer's comment. We agree with his/her suggestion. We have made the requested modifications.

Comment No 4: Line 167: Provide the longitude and latitude here.

Response: Thanks for the reviewer's comment. We agree with his/her suggestion. We have made the requested modifications.

Comment No 5: Line 177: Screening for Biosurfactant Production, please include related references to support the method.

Response: Thanks for the reviewer's comment. We agree with his/her suggestion. We have made the requested modifications.

Comment No 6: Results and discussion, where are the identification results? Please add detailed identification results here.  

Response: Thanks for the reviewer's comment. This information is mentioned in lines 265-290.

Comment No 7: Line 242: Table 1, use the three-dimensional diagram for the table. The same as other tables.

Response: Thanks for the reviewer's comment. However, we don’t find that necessary. Table 1 report that information about locations and species isolated. We don’t aim to compare them, unlike other tables.

Comment No 8: Line 363: Table 3, some parts are missing in the table.

Response: Thanks for the reviewer's comment. However, we couldn’t detect the missed parts in our version of the manuscript. Please review the current version.

Comment No 9: Line 371: Leahy et al. (2003), please check the format.

Response: Thanks for the reviewer's comment. We agree with his/her suggestion. We have made the requested modifications.

Comment No 10: Line 378-383: Fusarium strains are known for their metabolic capability to degrade various organic pollutants such as Biodegradation of allethrin by a novel fungus Fusarium proliferatum strain CF2, isolated from contaminated soils. Microorganisms, 2020. Why the authorS did not compare the degradation activity of F. verticillioides and F. proliferatum with that of other Fusarium strains based on the literature. Authors should add more information into this section and cite the recent research into the field.

Response: Thanks for the reviewer's comment. We agree with his/her suggestion. However, we had used many recent references to compare our results with them. Please review the following references (28, 46, 47, 48, 49, 53, 63, 64, 65, 77, 78, 79, 80, 81, and 82).

Comment No 11: Line 564-609: Authors should add more information into this section and cite the recent research into the field such as: Biosurfactant is a powerful tool for the bioremediation of heavy metals from contaminated soils. Journal of Hazardous Materials, 2021; Microbial glycoconjugates in organic pollutant bioremediation: recent advances and applications. Microbial Cell Factories, 2021. This way the authors will demonstrate that they really have a good knowledge of the related literature.

Response: Thanks for the reviewer's comment. We agree with his/her suggestion. We have made the requested modifications.

Comment No 12: Conclusion: This section should be revised for the better understanding of the topic and its future research.

Response: Thanks for the reviewer's comment. We agree with his/her suggestion. We have made the requested modifications.

Comment No 13: References: Many of the references have been superceded and more modern ones are required such as Microbiol. Rev. 1990, 54, 305–315; Biotechnol. Tech. 1993, 7, 745–748; Int. Biodeterior. Biodegradation. 2004, 54(1), 61-67..

Response: Thanks for the reviewer's comment. We agree with his/her suggestion. However, we mentioned both old and recent publications in the same paragraph. For instance, Reference no. 36 (1993) was coupled with 37 (2021) and reference 57 (1190) was coupled with reference 57 (2007). From our point of view, that strengthen our hypothesis and serve the aims of the study.

Reviewer 4 Report (New Reviewer)

In this manuscript, the authors discuss the biodegradation of contaminated soil samples using different species. Overall, the manuscript is very well written and the methods are explained well. However, the manuscript needs a minor revision to address some of the comments below

1. The authors do not justify the selection of the soil samples from specific location. Any reason why the soil samples were collected at these locations?

2. Please explain why Fusarium species was selected for this research? Is this a commonly occurring fungi species in SA?

3. In general, literature review needs to be expanded to provide some justification for this study.

4. Some of the methods are their purpose is unclear. For example: Oil Spreading Assay and Emulsification activity are not defined clearly. Authors seems to cite a  recent self-publication to explain this.

5. Similarly germination assay part of the methods (2.7) seems really unusual. It looks like the seeds were immersed and then rinsed with distilled water. How  does this simulate the germination process in the real soil sample? I am unsure if this method captures the real-world scenario of seed germination process.

6. In the results and discussion section, I am unsure how Figure 4 & 5 contribute to the author's hypothesis? Figure 4A, showing the healthy control, has several knots and bumps so I am not sure if this is different in B and C? The authors should explain this to the reader to clarify.

7. 478-480 lines seem to be saying that these deformities observed in SEM are due to the reaction with fatty acid content. I am not sure if this method can establish this conclusively. This line should be deleted or further clarification or citation should be provided

8. How are the biosafety levels evaluated in this study? I am unsure if germination of tomato seeds can be enough to say that these are safe for consumption.

Author Response

Reviewer No.4

Comment No. 1:  The authors do not justify the selection of the soil samples from specific location. Any reason why the soil samples were collected at these locations?

Response: Thanks for the reviewer's comment. We agree with his/her suggestion. To explain the criteria of locations selection, the reviewer will notice that all locations for soil samples are in gas stations in Riyadh, where oil hydrocarbons were collected from the oil pipelines of Aramco company in Dammam. We selected the soil sample locations in Riyadh, as this location is far enough from the primary source of Petroleum exploration sites and at the time it’s one of the most crowded cities in Saudi Arabia, so we expect the availability of microbial species will be higher and that is reported in Table 1. In contrast, oil samples for our study were collected from the main pipelines to overcome any other source of microbial contamination. We clarified this point in the results and discussion section.

Comment No. 2. Please explain why Fusarium species was selected for this research? Is this a commonly occurring fungi species in SA?

Response: Thanks for the reviewer's comment. The selection of Fusarium species depended on the results reported in Table 1. The table showed that their percentage is more than 26% in some locations. Also, the current study is one of the studied performed to screen all of theses species in separate settings. Please review the following publication:

Al-Zahrani, R.M.; Al-Otibi, F.; Marraiki, N.; Alharbi, R.I.; Aldehaish, H.A. Biodegradation of Petroleum Hydrocarbons by Drechsleraspicifera Isolated from Contaminated Soil in Riyadh, Saudi Arabia. Molecules 202227, 6450. https://doi.org/10.3390/molecules27196450

Comment No. 3. In general, literature review needs to be expanded to provide some justification for this study.

Response: Thanks for the reviewer's comment. We agree with his/her suggestion. However, we used 82 references to compare our results with different findings. we don’t think the journal might allow more citations.

Comment No. 4. Some of the methods are their purpose is unclear. For example: Oil Spreading Assay and Emulsification activity are not defined clearly. Authors seems to cite a  recent self-publication to explain this.

Response: Thanks for the reviewer's comment. We agree with his/her suggestion. However, this information is mentioned all over the manuscript, for example sub-section 2.5. (Screening for Biosurfactant Production) in which we used three methods entitles; Drop Collapse assay, Oil Spreading Assay, and Emulsification Activity, so it’s clear they were used to prove the production of biosurfactants as they will increase the surface tension and affects the characteristics of tested oils. Other modifications were performed in the results and discussion section to clarify that point.

Comment No. 5. Similarly germination assay part of the methods (2.7) seems really unusual. It looks like the seeds were immersed and then rinsed with distilled water. How  does this simulate the germination process in the real soil sample? I am unsure if this method captures the real-world scenario of seed germination process.

Response: Thanks for the reviewer's comment. We don’t agree with his/her suggestion. The germination assay is one of the techniques used to assess the sustainability of microbial biosurfactants and their effect on the germination. Please review references 40 and 81.

  1. In the results and discussion section, I am unsure how Figure 4 & 5 contribute to the author's hypothesis? Figure 4A, showing the healthy control, has several knots and bumps so I am not sure if this is different in B and C? The authors should explain this to the reader to clarify.

Response: Thanks for the reviewer's comment. Here we discuss the ultrastructural changes inby SEM imaging. As been described in the manuscript figure 4 showed that the treatment with 1% crude oil caused swellings and blurring edges at the knots, where spores had irregular surfaces and edges, which is in contrast to the control. Similarly, figure 5 revealed swelling, blurring, and knots forming on the outmost edges, which decreased the smoothing of the cellular walls and the microconidia structures were not clear or disappeared. We think it’s clearly discussed.

  1. 478-480 lines seem to be saying that these deformities observed in SEM are due to the reaction with fatty acid content. I am not sure if this method can establish this conclusively. This line should be deleted or further clarification or citation should be provided

Response: Thanks for the reviewer's comment. We agree with his/her suggestion. We removed these lines.

  1. How are the biosafety levels evaluated in this study? I am unsure if germination of tomato seeds can be enough to say that these are safe for consumption.,

Response: Thanks for the reviewer's comment. We don’t agree with his/her suggestion. The germination assay is one of the techniques used to assess the sustainability of microbial biosurfactants and their effect on the germination. Please review references 40 and 81.

Round 2

Reviewer 3 Report (New Reviewer)

The manuscript was rigorously evaluated by several reviewers, who provided contributions and the authors accepted the suggestions. The manuscript can be accepted for publication according to the revisions carried out.

Author Response

Reviewer comment: The manuscript was rigorously evaluated by several reviewers, who provided contributions and the authors accepted the suggestions. The manuscript can be accepted for publication according to the revisions carried out.

Response: Thanks

This manuscript is a resubmission of an earlier submission. The following is a list of the peer review reports and author responses from that submission.

Round 1

Reviewer 1 Report

The manuscripts described the isolation and characterization of fungi (Fusarium spp.) showing hydrocarbon (oils) tolerance. Overall, the manuscript is well written. However, the present data just showed the basic results. There is not much data explain for the reasons why the isolates have hydrocarbon tolerance, making the manuscript lacks the novelty. I would recommend the authors pay attention to the following issues:

1. The fungi are described based on morphology. Although the description indicated the isolates are belong to Fusarium genus. It would be more confident if the authors perform molecular identification such as analysis the ITS1-4 sequence.

2. The 2, 6-Dichlorophenol Indophenol (DCPIP) assay is measured based on the loss of the color of DCPIP (blue to white). Some articles demonstrated the highest absorbance wavelength of DCPIP reduced form is 600 nm (doi.org/10.1007/s00775-020-01752-9). Meanwhile, the cited reference did not provide the wavelength clearly. Thus, the author must check the assay again. Moreover, the loss of color will lead to the reducing of absorbance value. However, the data presented in figure 7 showing the increasing of absorbance values. I don’t know how it happened.

3. Fusarium spp. are well known with hydrocarbon tolerance abilities due to capable to produce enzyme such as oxidases and reductases. The authors also discussed about this (L337 to L344). However, none of these enzymes have been assayed. The authors should measure the dynamic of some enzyme activity during treatments such as laccase (one of the main peroxidases released by Fusarium). This could explain for the tolerance mechanism of oils by Fusarium.

4. The data showing tables are not statistical analysis as the authors mentioned in materials and methods section. Please check again.

5. As same as the role of enzyme, biosurfactants contribute to oils tolerance. This would indicate the tolerance may relate to the amount of releasing biosurfactants. The table 6 showed that the recovery of biosurfactants produced by Fusarium spp. which were growth in medium containing crude oils higher than that in mixed oils. In contract, the table 4 and 5 indicated drop collapse and emulsification activity of CFS (contained biosurfactnats) produced in medium containing in mixed oils were higher in compared with crude oils. The author must explain for it.

Other comments:

Name of microorganism must be written in italic.

Check again the terms Fusarium sp. (singular) and Fusarium spp. (plural).

Check the format of using references

Reviewer 2 Report

Paper entitled “Biodegradation of selected hydrocarbons by Fusarium species isolated from contaminated soil samples in Saudi Arabia” describe the efficacy of three Fusarium species isolated from contaminated sites to degrade the different hydrocarbons. The authors used SEM to investigate the morphological changes of treated fungi. Also, the biodegradation efficacy was checked by 2, 6-Dichlorophenol Indophenol (DCPIP), drop collapse, emulsification activity, and oil Spreading assays. The manuscript contains promising results and is well-arranged. However. The manuscript needs major revision before being accepted for publication in JOF.  

1-    The result part in the abstract should be rephrased to contain the promising data

2-    Line 44, “Bioremediation of degradation of oil….” Should be “Bioremediation of oil….”

3-    Please refer to the novelty of the current study in the introduction section.

4-    The introduction section should refer to the efficacy of different fungal strains in the biodegradation of various contaminants. I recommend the following to cite: https://doi.org/10.3390/jof7030193; https://doi.org/10.21608/EJCHEM.2019.11720.1747

5-    Line 54, 83, 192 “Fusarium sp.” should be “Fusarium sp.”. The scientific name must be in italics throughout the manuscript, please check and correct it.

6-    Line 56, “screaming” I think it should be “screening”. Please check and revise.

7-    The hypothesis of the current study “Line 54 – 59” should be rephrased to concise and clear.

8-    Line 62, “oil samples” or “soil samples”, please check and revise.

9-    Line 62 – 64, Please add the coordinate for the collected oil samples used for fungal isolation.

10- Line 67, “PH” should be “pH”, please revise throughout the manuscript.

11- Line 69, “(Dammam, Saudi Arabia)” adding coordinates.

12- Line 85, “Pictorial atlas of Watanabe, (2010) [23].” Delete (2010).

13- Please standardize all units throughout the manuscript such as “ml” “mL” “min.” mins”, and so on.

14- Line 120, “1 cm2” should be “1 cm2

15- Line 232 and 233, please check the scientific names to be italics.

16- Please add the culture and reverse photo and microscopic examination of three Fusarium isolates

17- Authors isolated different fungal species as shown in Table 1, whereas during screening on the ability of these fungal to tolerate oil on solid media use only Fusarium sp. why? It should be screening all isolates, it may isolate other than Fusarium and give more activity, please clarify.

18- The values on the column in Figure 2 A are overloaded, please check and revise.

19- What about the optimization study such as temperature, pH another carbon source, nitrogen source, and inoculum size,… these parameters have critical roles in biodegradation

20- The seed germination figure 9 is not clear, authors can select one or two seedlings to photo.

21- The germination rate due to F. oxysporum and F. verticillioides decreased after 18 days, who? Please clarify.

22- The conclusion should be rephrased to show the advantages and limitations of the current study.

23- Authors should revise the manuscript carefully to correct the typo- and grammatical errors.  

Reviewer 3 Report

In the last decades, biodeterioration has gained great interest in the light of sustainable and cost-effective bioremediation. The manuscript falls in this area, but it is affected by so many concerns (formal and methodological) that are impossible to consider for a possible publication.

Beyond the problems tied to the language (an English revision is due), the study design is confusing, the speech is fragmentary and incomplete (especially in the material and methods section), discussion and conclusions are not supported. The authors should first decide what they want to investigate. If they want to prepare a manuscript focused on methods, biosurfactants, fungal eco-physiological responses to xenobiotics or on their biodegradation. In this sense, there is a discrepancy between the title and the manuscript contents. In detail, the authors claim a biodeterioration study, but they perform some (1) experiments to measure tolerance against xenobiotics. Tolerance is not biodegradation. They describe the (2) morphological changes occurring on fungi exposed to hydrocarbon by SEM, but morpho changes are not a way to measure and characterize biodegradation. (3) they measure biosurfactant production using four different methods, posing a concern about the most reliable method. And unfortunately, this is not a method evaluation research. (4) the germination trial as those to characterize the surfactants produced are tests not relevant to describe the biodegradation of the considered pollutants.

Grave concerns are also tied to the method used to isolate fungal strains from contaminated soils and fungal identification. Indeed, the medium used was potato dextrose agar, a generic medium rich in dextrose (20g/L) in which whatever fungus and even some bacteria could grow. Otherwise, it is common practice to use a selective culture medium (sometimes containing mid-low concentrations of the xenobiotic we want to degrade) to save time and perform a reliable selection of strains.

The identification of strains has been performed using morphological keys (dated back to 2010), and strains have no collection number nor deposited barcode sequences. This means that the identifications are unreliable (molecular identification must be performed using more than one barcode region), and the experiment has no traceability. The selection of Fusarium strains has not been justified. Results on biodegradative abilities are banal (measured by increasing biomass and not by substrate degradation) since it is well known that aliphatic hydrocarbons are easily metabolized by a wide range of microorganisms evidenced even by the tons of papers reporting the crude oil and/or derived fuels fouling.

 More important, this manuscript is a carbon copy of a preprint having as subject Candida species (https://assets.researchsquare.com/files/rs-1446607/v1/c089482d-8a78-4e6a-82a2-84023b93d5f3.pdf?c=1647542216) and an already published paper on Molecules having, this time, as subject Drechslera (https://www.mdpi.com/1420-3049/27/19/6450/htm).

Below are some additional notes.

Abstract – bioremediation (L9) and biodegradation (title) are not synonyms. Indeed, bioremediation includes all methods that reduce the presence of environmental pollutants (i.e. adsorption). In any case, the use of microorganisms is preferred to others, not for safety (all methods must be safe) but because they are cost-effective, especially if the degradation is complete. Please amend this item along with the text.

L9, L43, L45 etc not safest but sustainable / cost effective

Keywords should be changed because they are repeats of the words of the title. Please take a look at the link below (but there are many others) to have some suggestions for choosing the most effective keywords. https://scientific-publishing.webshop.elsevier.com/manuscript-preparation/how-choose-keywords-manuscript/

Introduction. The background information is not complete. No mention of the differences /composition occurring in the hydrocarbon mixtures they want to degrade (e.g. diesel, kerosene, used oil, and mixed oil) and problems tied to their degradation.

L27-28 The sentence contains a number of errors: hydrocarbons are obviously rich in carbon as the name suggest. More serious is the error in basic chemistry where alkanes and cycloalkanes are suggested to be aromatic compounds (e.g. alkanes, cycloalkanes and other aromatic compounds).

L39…it is necessary to understand the properties of the crude oil – the properties of crude oil have been extensively studied.

L44 Bioremediation of degradation of – please correct

L47 …has been explored in myriad research studies…As the authors stated, the biodegradation of hydrocarbon pollutants has been extensively studied. For this reason, is of utmost importance to highlight the novelty of this research and what would be the gain in knowledge given by this study. For this reason, I suggest being more precise in reporting the species involved and the hydrocarbon treated.

L54 investigated the biodegradation of three Fusarium….please modify because the title suggests the hydrocarbon biodegradation, but here is Fusarium the biodegradation object.

Materials and methods (M&M) miss the basic information necessary to reproduce the experiment and give reliability to the results.  The test design here reported is puzzling; it misses linearity and, as before, is not consistent with the title and research purpose.

2.1 Soil and hydrocarbon collection. The order in which oil (L62) and soil (L65) sampling are reported is not congruent with the title. Moreover, the coordinates of the oil sampling sites should be included. No mention of the soil sampling sites. No mention of the used oils.

L65 sterilized containers- what kind of containers?

L67 pH instead of PH

L69 from oil tankers of Aramco Co. As the crude-oil composition and their derivates could change, their composition (at least in the major components should be given).

L71 Please better define the composition of mixed oils.

L72 in sterile bottles. What kind of bottles?

2.2 Characterization and identification of the fungal strains

The section header is not consistent with its contents. The isolation protocol has been described, and no characterization descriptions have been reported. For this reason, the header should be changed. “Isolation and identification…” fits best with the reported contents.

L74 The sentence should be reworked if not cancelled. It is clearly a self-citation, and the sentence as it is does not give peculiar information.

L75 sterile distilled water, please modify.

L81 until the experimental day, please modify

L83-88 this section is confusing because the Fusarium species that are reported in the result section are also here reported. This is nonsense because if the species are here reported, the whole section of sampling, culture and identification should be removed. Otherwise, the morphological identification cannot be accepted because the genus taxonomy has been modified after 2010, and it is well known that the morphological identification cannot be accepted, especially in wide groups such as Fusarium, Cladosporium, Aspergillus etc Molecular identification by sequencing is required (using more than one barcode region).

2.3 growth of fungi on solid media

This header should be changed because even PDA is a solid media, and experiments focused on tolerance trials.

L90 Mineral salt medium (MSM) agar medium. “medium” is repeated please modified.

L90 1% please specify w/v, w/w etc.

Please define the amount of culture medium poured into each plate and information about the inoculum.

2.4 growth in liquid media

Please describe the instrument used to weigh the flasks every three days  (and its precision) and the number of repeated measures for each sample. In this experiment, controls have no carbon source, that’s an easy win. L104 grams is g.

2.6. DCPIP Assay

DCPIP assay is an indicator of metabolic activity. In this section, there is no mention of controls. To set best the experiment is important to know that (1) fungi, even when carbon sources are not present, maintain metabolic activity for a while and produce inspective hyphae, and (2) DCPIP could be subject to light decay. No mention of experiment replicas

L120  cm squared, change 2 to superscript

Result and discussion

When data are achieved using different methods, creating different sub-sections with relative titles is of utmost importance.  All sentences reporting obvious statements, such as at L271 "the growth rate of Fusarium differed depending of the carbon source used"  should be removed.

L183-186 is a mix of materials and methods and data out of the research aim (the frequency of investigated fungi and the properties of soil samples…Table 1). Please discard.

L192-221 should be discarded for the abovementioned reasons.

Figure 3D misses the control. In any case, a little growth of controls is expected.

Table 3. because controls started from significantly different biomasses, the results achieved are questionable.

L334-337 The sentence is too general in any case, Aspergillus is reported twice, and authors do not assess the degradation of whatever chemical. So, even the following sentences are not supported statements.

L345-439 As before SEM observation have no scientific relevance. Remove

The rest of the manuscript is affected by the abovementioned problems.

Other notes.

Figures and tables should be considered stand-alone material. For this reason, their captions should report all information necessary for understanding, such as units, acronyms etc.

Be consistent in reporting genus and species names. All of them should always be in italics; species names should not be capitalized

Be consistent in reporting the titles of papers cited; some are capitalized, and others aren't. Capitalized titles are not required in the reference list.

Suggestion Citing papers published by journals indexed with low scores (Q) or not indexed at all doesn’t give a good impression.

Self-citation if accepted if justified. “, as described before” in the M&M section, it is not (L150).

L327 There is a citation without a number